# Global dataset on seagrass meadow structure, biomass, production and reproduction

Simone Strydom[1,2], Chanelle L. Webster[1], Caitlyn M. O'Dea[1], Nicole E. Said[1], Roisin McCallum[1], Karina Inostroza[3], Cristian Salinas[1], Samuel Billinghurst[1], Anna Lafratta[1], Charlie M. Phelps[1], Connor
Campbell[1], Connor Gorham[1], Natasha Dunham[1], Rachele Bernasconi[1], Anna M. Frouws[1], Axel Werner[1], Federico Vitelli[1], Viena Puigcorbé[1,4], Alexandra D'Cruz[1], Kathryn M. McMahon[1], Jack Robinson[1], Megan J. Huggett[5], Sian McNamara[1], Glenn A. Hyndes[1] and Oscar Serrano[1,6*]

[1]Centre for Marine Ecosystems Research and School of Science, Edith Cowan University, Joondalup, 270
Joondalup Dr, WA 6027, Australia.

[2]Marine Science Program, Biodiversity and Conservation Science, Department of Biodiversity, Conservation and Attractions, 17 Dick Perry Ave, Kensington, WA 6151, Australia.

[3]Biosfera, Associació d'Educació Ambiental, Catalonia, Spain

[4]Institut de Ciències del Mar, Consejo Superior de Investigaciones Científicas, 08003 Barcelona, Spain.

[5]School of Environmental and Life Sciences, The University of Newcastle, 10 Chittaway Rd, Ourimbah, NSW 2258, Australia.

[6]Centro de Estudios Avanzados de Blanes, Consejo Superior de Investigaciones Científicas, Blanes, Spain.

*Correspondence to: Oscar Serrano (oserrano@ceab.csic.es)

**Abstract.** Seagrass meadows provide valuable socio-ecological ecosystem services, including a key role in climate change
mitigation and adaption. Understanding the natural history of seagrass meadows across environmental gradients is crucial to decipher the role of seagrasses in the global ocean. In this data collation, spatial and temporal patterns in seagrass meadow structure, biomass, production and reproduction data are presented as a function of biotic and abiotic habitat characteristics. The biological traits compiled include measures of meadow structure (e.g., percent cover and shoot density), biomass (e.g., above-ground biomass), production (e.g., shoot production), and reproduction effort (e.g., flowering intensity and seed bank
density). Categorical factors include bioregion, geotype (coastal or estuarine), genera and year of sampling. This dataset contains data extracted from peer-reviewed publications published between 1975 and 2020 based on a Web of Science search, and includes 15 data variables across 12 seagrass genera. The top four most studied genera are *Zostera*, *Thalassia*, *Halophila* and *Cymodocea* (80% of data), and the least studied genera are *Phyllospadix*, *Amphibolis* and *Thalassodendron* (2.3% of data). The data hotspot bioregion is the Tropical Indo Pacific (25% of data), whereas data for the other five bioregions are evenly
spread (ranging between 13 and 16% of total data within each bioregion). From the data compiled, 39% related to seagrass biomass, while the least number of data were related to seagrass production (10% of data). This data collation can inform several research fields beyond seagrass ecology, such as the development of nature-based solutions for climate change mitigation, which include readership interested in blue carbon, engineering, fisheries, global change, conservation and policy.





## 1 Introduction

Approximately 65 million years ago, a group of marine angiosperms called seagrasses adapted to life within the coastal zone, and now, they rank among the most valuable ecosystems globally. Seagrasses encompass ~72 species within 12 genera spread across all continents except Antarctica (Short et al., 2011). Seagrasses are recognised as highly productive habitats that provide multiple ecosystem services relevant to human wellbeing, such as biodiversity, fisheries, sediment stabilisation and nutrient cycling across the coastal zone (McMahon et al., 2013; Nordlund et al., 2016; Unsworth et al., 2018). Furthermore, the high primary production rates and capacity of seagrasses to sequester carbon is relevant to mitigating climate change, while their role in stabilising the substrate, ameliorating hydrodynamic energy and nourishing beaches with biogenic sands contributes to climate change adaptation against storms and sea-level rise (Duarte et al., 2013).

Seagrass research initially focussed on understanding biology, distribution, ecology, taxonomy, and phenology. More recently, the socio-ecological value of seagrass ecosystem services has received recognition, in part owing to the extensive losses of seagrasses globally. Since the beginning of the 20th century, widespread loss of seagrass meadows has been estimated at 0.9% yr[-1], linked to a variety of factors including impacts associated with alterations to key drivers of growth (e.g., irradiance and temperature) resulting from sediment loading, eutrophication, extreme climate events and flooding (Hall et al., 1999; Short et al., 2011; Strydom et al., 2020; Waycott et al., 2009). Recent conservation and management actions have resulted in the deceleration and reversal of declining trends in some locations (de los Santos et al., 2019).

Duarte and Chiscano (1999) conducted a review on seagrass biomass and production, which has greatly contributed to the advancement of seagrass research. Information on seagrass meadows structure, production, biomass and reproduction is essential to understand the role of seagrasses in the global ocean, while providing insights for developing restoration initiatives, informing management and ultimately contributing to their conservation (Unsworth et al., 2018). Therefore, understanding global patterns in the functioning of threatened natural ecosystems such as seagrass meadows, is crucial to inform management strategies to protect natural assets (Cullen-Unsworth et al., 2014; Hoegh-Guldberg and Bruno, 2010). Since Duarte and Chiscano (1999), new information across hundreds of peer-reviewed manuscripts (past 24 years, 1996-2020) has not been synthesised and made available online, which precludes gathering new knowledge around seagrass natural history based on data synthesis studies. Indeed, data on seagrass reproduction has never been compiled.

In this review, data on key variables on seagrass meadow structure, biomass, production, and reproduction published between 1975 and 2020 (data collected between 1964 and 2019) are presented as a function of biotic and abiotic habitat characteristics. The main goals of this review are to synthetize current literature on seagrass ecology to facilitate further multidisciplinary research, and to identify research gaps and provide recommendations for future research. The dataset provides baseline data that can inform science, management and policy. In particular, it provides critical and basic knowledge to inform traditional seagrass biology and ecology fields, but also can contribute to advance knowledge in other disciplines including fisheries, biodiversity, conservation, coastal biogeochemistry, and emerging fields such as the Blue Economy.



## 2 Data compilation

### 2.1 Literature search

In order to create a global seagrass database containing relevant data on seagrass meadow structure, biomass, production, and reproduction, a Web of Science (www.webofknowledge.com) search was conducted in June 2020 using these search terms for the growth, production and biomass variables:

((TS=((Seagrass* OR eelgrass OR SAV OR Amphibolis OR Cymodocea OR Enhalus OR Halophila OR Halodule OR Posidonia OR Phyllospadix OR Ruppia OR Thalassia OR Thalassodendron OR Zostera) AND (product* OR biomass OR growth OR exten) )))

Then another search for reproduction variables using these terms: ((TS=((Seagrass* OR eelgrass OR SAV OR Amphibolis OR Cymodocea OR Enhalus OR Halophila OR Halodule OR Posidonia OR Phyllospadix OR Ruppia OR Thalassia OR Thalassodendron OR Zostera) AND (germinat* OR reprod* OR seed* OR flower* OR fruit* OR dispersal * OR gra$ing*)).

Only data from peer-reviewed manuscripts was included and thereby, the dataset compiled excludes data from non-peer reviewed manuscripts and reports. Data for 15 variables (mean values) were extracted (Table 1, see 2.2) and compiled in a database (https://doi.org/10.1594/PANGAEA.929968). These variables were selected based on their widespread study among seagrass habitats, and to their usefulness for quantifying seagrass condition across papers with different aims (i.e., monitoring condition vs reproductive effort) (Short & Coles, 2001). Standardised units (spatially i.e., $m^{-2}$ and temporally i.e., $day^{-1}$) are reported. Note that data from mesocosm experiments, field experiments with the exception of control sites, or meadows altered by direct anthropogenic disturbances (i.e., aquaculture, anchoring or dredging) were dismissed as these were considered as impacted meadows and were unlikely to reflect meadows in a 'natural' condition. Note that seagrass species were recorded following accepted convention as per Short et al. 2011 (e.g., *Zostera capricorni*, *Z. mucronata*, *Z. novazealandica* were named as *Z. muelleri*).

### 2.2. Seagrass structure, biomass, production and reproduction variables

The 15 variables extracted from the peer-reviewed literature were classified within four categories: seagrass meadow structure (3 variables), biomass (3 variables), production (5 variables) and reproduction (4 variables).

- Seagrass meadow structure: percent cover (%), shoot density (no. shoots $m^{-2}$), leaf density (no. leaves $m^{-2}$).
- Seagrass biomass: above-ground biomass dry weight (DW) (g DW $m^{-2}$), below-ground biomass (g DW $m^{-2}$) and total biomass (g DW $m^{-2}$).
- Seagrass production: shoot production (g DW $m^{-2}$ $day^{-1}$), leaf production (g DW $m^{-2}$ $day^{-1}$), and above-ground production (g DW $m^{-2}$ $day^{-1}$), below-ground production (g DW $m^{-2}$ $day^{-1}$) and total production (g DW $m^{-2}$ $day^{-1}$).
- Seagrass reproduction: flowering intensity or flowering shoots (no. flowers $m^{-2}$), fruit density (no. fruits $m^{-2}$), seed bank density (no. seeds $m^{-2}$) and seedling density (no. seedlings $m^{-2}$).

For all these 15 variables, relevant data points were extracted from results text, tables and when appropriate from figures using a web based tool that allow the extraction of data from plots, images and maps (WebPlotDigitazer: https://automeris.io/WebPlotDigitizer/). Datapoint in an individual row can be a mean of multiple replicates or a single unique measurement point for that variable and location. Other relevant spatial and site information was also extracted including the latitude and longitude (decimal degrees), seagrass bioregion according to Short et al. 2007 (Fig. 1), geotype (coastal or estuarine geomorphology), seagrass genera, the year of sampling when reported, and the doi of the publication containing the





data. When sampling site coordinates were not reported in the publication, study location maps were consulted if applicable and corresponding coordinates estimated using *Google Earth*. Similarly, geotype was classified as estuarine if the study site was on close proximity to riverine input or within a coastal lagoon, conversely if there were no rivers nearby or the study site
was located within an embayment then it was considered coastal. For the flowering intensity variable, reproductive shoots were included in this dataset variable (i.e., studies on *Ruppia* counted reproductive shoots and as these had flowers on them, they were considered an analogous term). Furthermore, if flowers were identified as male or female in studies, they were included in the dataset as total number of flowers per $m^{-2}$ regardless of gender. Indeed, details on density of flowers, spathes, inflorescence shoots and reproductive shoots where combined into a single variable (i.e., flowering intensity). If publications
included data on above-ground biomass and below-ground biomass for the same study site, these two values were summed to estimate a value of total seagrass biomass. Publications that reported growth or production expressed as grams of carbon were excluded. When sampling was conducted over multiple years, the year of sampling was left blank and not reported in the dataset.

The seagrass natural history information reported and the way it was reported has evolved during the 45 years of research compiled. Overall, early publications provided comprehensive details regarding the description of flowers, seeds and fruits, while sampling procedures were not clearly described. Later on, the sampling strategies and data reporting became more standardized and comprehensive.

### 2.3 Statistical analyses

Descriptive parameters (e.g., count of data and publications, minimum, maximum and median values) for all 15 variables were compiled. Median values are reported instead of mean values because the data for most of the variables studies is not normally distributed. Boxplots for four key variables sorted by bioregion and genera were produced in R using the ggplot2 package (Wickham, 2016) (version 4.0.1, R Core Team 2020). In order to spatially illustrate the dataset, maps were also created in R, using the leaflet package (Graul, 2016).

## 3 Results and discussion

The highest number of data points were collected in year 2016, while the lowest occurred in 1969 (Fig. 2). Overall, all four data categories were represented well over time (1964–2019), with biomass data present in the majority of papers consistently over time, meadow structure data encompassing a larger proportion of data over the last decade, and reproduction and production data being the least studied categories. Approximately 59% of the studies were conducted in coastal marine areas
(n = 3,302) with the remaining 41% of studies conducted in estuarine areas (n = 2,285).

### 3.2 Spatial distribution of seagrass data
The seagrass database includes information collected across 15 variables on seagrass structure, biomass, production and reproduction from all 12 seagrass genera described to date, spanning all continents except Antarctica (Fig. 3). Based on the count of data, the top five most studied genera making up to 80% of the database were *Zostera* (n = 5,511), *Thalassia* (n =
1,351), *Halophila* (n = 1,266), and *Cymodocea* (n = 1,241). The least studied genera were *Amphibolis* (n = 58), *Thalassodendron* (n = 87), and *Phyllospadix* (n = 126). The predominance of *Zostera* data could be related to their broad global distribution, including European countries which were the pioneers of seagrass science, while the least studied genera are more geographically restricted (Fig. 3). The bioregion with highest number of data was the Tropical Indo Pacific (n = 2,950), which also included 10 of the 12 genera, illustrating the seagrass biodiversity of this bioregion. The number of data
across the Temperate North Pacific (n = 1,911; 4 genera), Mediterranean (n = 1,905; 5 genera), Temperate North Atlantic (n





= 1,809; 3 genera), Temperate Southern (n = 1,634; 5 genera), and Tropical Atlantic (n = 1,564; 6 genera) bioregions was similar. There was up to 95-fold difference between the most and least studied seagrass genera, but only a 2-fold difference between bioregions. The most prevalent data type was seagrass biomass (n = 6,087; 52%), followed by structure (n = 3,256; 28%), reproduction (n = 1,181; 10%) and production (n = 1,249; 11%) (Fig. 4).


### 3.3 Variability in seagrass data among variables

The dataset compiled includes data on shoot density (n = 2,366), percent cover (n = 731), leaf density (n = 159), above-ground biomass, (n = 2,519), below-ground biomass, (n = 1,488), total biomass (n = 2,080), shoot production (n = 110), leaf production (n = 670), above-ground production (n = 192), below-ground production (n = 89), total production (n = 188), flowering

intensity (n = 706), fruit density (n = 55), seed bank density (n = 312) and seedling density (n = 108). Overall, production was the least reported variable type (n = 1,249), followed by reproduction (n = 1,181). Seagrass structure and biomass variable types were the most reported (n = 3,256 and 6,807, respectively). Across all dataset, shoot density ranged from 0.08 to 28,682 shoots m$^{-2}$ (median = 651), percent cover from 0.03 to 100% (median = 35.2), leaf density from 5.1 to 48,978 leaves m$^{-2}$ (median = 3,287), above-ground biomass from 0.0010 to 1,509 g DW m$^{-2}$ (median = 53.2), below-ground biomass from 0.0340

to 3,076 g DW m$^{-2}$ (median = 64.0), total biomass from 0.0010 to 3,393 g DW m$^{-2}$ (median = 148), shoot production from 0.0006 to 23.5 g DW m$^{-2}$ day$^{-1}$ (median = 2.44), leaf production from 0.0012 to 277 g DW m$^{-2}$ day$^{-1}$ (median = 1.14), above-ground production from 0.0003 to 23.5 g DW m$^{-2}$ day$^{-1}$ (median = 1.55), below-ground production from 0.019 to 34 g DW m$^{-2}$ day$^{-1}$ (median = 2.20), total production from 0.0018 to 38.5 g DW m$^{-2}$ day$^{-1}$ (median = 3.50), flowering intensity from 0.10 to 6,000 flower m$^{-2}$ (median = 16.1), fruit density from 0.5 to 3,229 fruits m$^{-2}$ (median = 142), seed bank density from 2.7 to

10,028 seeds m$^{-2}$ (median = 138), and seedling density from 0.001 to 7,560 seedlings m$^{-2}$ (median = 20.9).

There was high variability in most variables using pooled data across bioregions and genera, and in the amount of data for each variable across bioregion, geotype and genera (Table 1). The values of some variables varied substantially across the six bioregions (Fig. 5). Median total biomass was highest in the Mediterranean bioregion (269 g DW m$^{-2}$), while the lowest was in the Temperate North Atlantic bioregion (109 g DW m$^{-2}$). The highest median shoot density values were recorded in the

Temperate North Atlantic (1,606 shoots m$^{-2}$) and the lowest in the Temperate North Pacific bioregion (279 shoots m$^{-2}$). The highest median total production values recorded in the Temperate Southern bioregion (9.3 g DW m$^{-2}$ day$^{-1}$), while the highest median flowering intensity values were recorded in the Mediterranean bioregion (90 flowers m$^{-2}$). Of all genera, median total biomass was highest for seagrasses with persistent life history stages (Kilminster et al., 2015). *Posidonia* had the highest median total biomass (2,013 g DW m$^{-2}$), followed by *Phyllospadix* (1,055 g DW m$^{-2}$) (Fig. 6). Median shoot density values

were highest for *Phyllospadix* (6,593 shoots m$^{-2}$) followed by *Ruppia* (4,314 shoots m$^{-2}$). Total production was highest for *Phyllospadix* (median 22.3 g DW m$^{-2}$ day$^{-1}$), followed by *Syringodium* (median 9.3 g DW m$^{-2}$ day$^{-1}$). The highest median flowering intensity was recorded for *Syringodium* (1,983 flowers m$^{-2}$), followed by *Ruppia* (765 flowers m$^{-2}$) and *Halophila* (600 flowers m$^{-2}$).

### 3.4 Significant gaps

This global collation of seagrass data has illustrated some gaps in our collective peer-reviewed knowledge. Across seagrass' worldwide distribution, limited peer-reviewed data were found for the eastern Mediterranean, and the coastlines of South America and Africa. Data for some seagrass variables were spatially depauperate, such as seagrass production at high latitudes (<50°N and S), including the Temperate North Atlantic. Overall, production was the least reported variable type followed by

reproduction. When considering data among seagrass genera, the least studied were *Amphibolis* (n = 58), *Thalassodendron* (n = 87), and *Phyllospadix* (n = 126), with gaps in most variables. There was also a lack of reproductive information for



*Amphibolis*, *Phyllospadix* and *Thalassodendron*. Lastly, there was no peer-reviewed published data found for production of *Ruppia*.

## 4 Conclusions

This database encompassing peer-reviewed data collected over the last 58 years provides an overview of seagrass distribution, biomass, production, structure and reproduction on a global scale. The top four most prevalent studied genera encompassing 80% of data were *Zostera* (mostly from the Temperate North Pacific), *Thalassia* (Tropical Atlantic), *Halophila* and *Cymodocea* (Tropical Indo Pacific and Mediterranean), and the least studied genera *Amphibolis*, *Thalassodendron Phyllospadix* (2.3% of data). Data hotspots include the Tropical Indo Pacific bioregion (25% of dataset; from 89 unique publications), whereas the

Tropical Atlantic bioregion had the least amount of data (13% of data; 79 publications). The strengths on seagrass natural history knowledge focus on seagrass biomass (54% of data), while the least number of data was related to seagrass reproduction (9% of data). Our review can inform several research fields beyond seagrass ecology, such as the development of Nature-Based Solutions for climate change mitigation and adaptation and Blue Economy, which include readership interested in blue carbon, engineering, fisheries, global change, conservation and policy.

## 5 Data availability

Data archived in the data repository PANGAEA (https://www.pangaea.de/tok/260c83db9a3b9396b3122a0f22de9a7109603e8d) (Strydom et al., 2022)

## 6 Code availability

R scripts used to generate figures and maps can be found in the supplement.

## 7 Team list

Simone Strydom, Chanelle L. Webster, Caitlyn M. O'Dea, Nicole E. Said, Roisin McCallum, Karina Inostroza, Cristian Salinas, Samuel Billinghurst, Anna Lafratta, Charlie M. Phelps, Connor Campbell, Connor Gorham, Natasha Dunham, Rachele Bernasconi, Anna M. Frouws, Axel Werner, Federico Vitelli, Viena Puigcorbé, Alexandra D'Cruz, Kathryn M. McMahon, Jack Robinson, Megan J. Huggett, Glenn A. Hyndes and Oscar Serrano.

## 8 Author contribution

OS conceived the idea. SS and OS lead the project, curated data and wrote the paper. SS, JR, SM, RB, MJH and OS conducted literature searches. SS, RM, KI, CW, NS, CS, SB, AL, CO, CC, CG, CMP, ND, AW, AF, RB, SM, FV, VP, AD, KM, JR, MJH, GH and OS contributed to the manuscript and/or extracted data from papers. SS, CW, RM, CMP and OS wrote scripts and created figures and tables. All authors reviewed the manuscript.



## 9 Competing interests

The authors declare that they have no conflicting interest.

## 10 Disclaimer

## 11 Acknowledgements

A.L. was supported by Environmental Protection Authority (SA)/Water SA and Edith Cowan University (ECU) Industry Engagement Scholarship. C.S. was funded by ECU Higher Degree by Research Scholarship. OS was supported by I+D+i projects RYC2019-027073-I and PIE HOLOCENO 20213AT014 funded by MCIN/AEI/10.13039/501100011033 and FEDER. We would also like to acknowledge the help of Madison Williams-Hoffman, Casper Avenant and Paul Lavery for their respective input into earlier versions of this manuscript, and the Centre for Marine Ecosystems research.

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

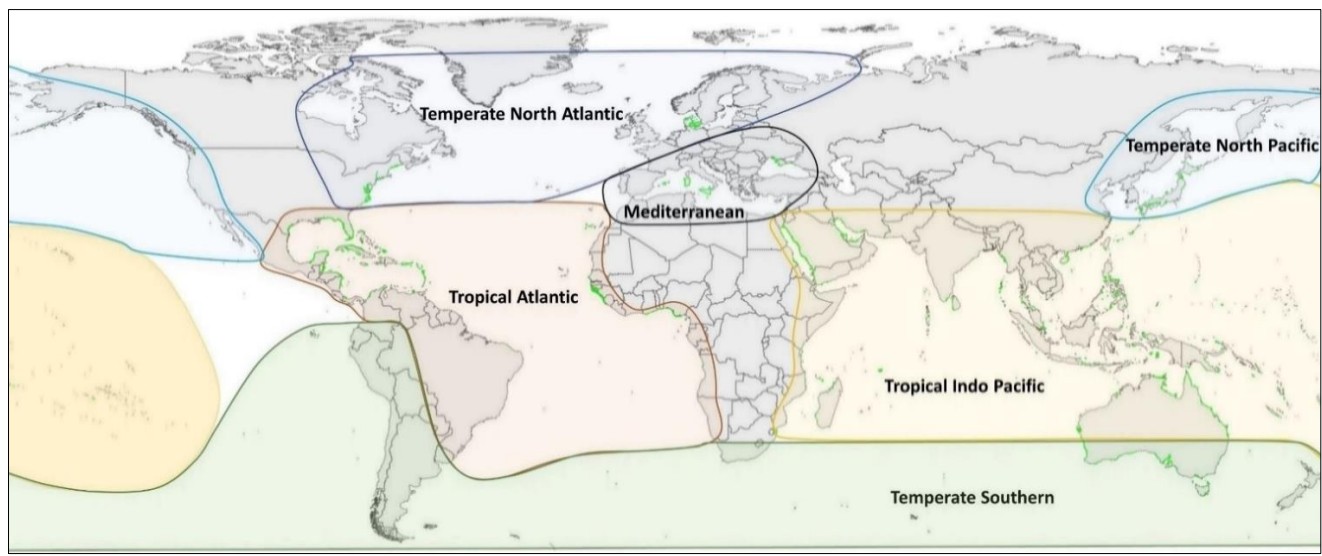

**Figure 1.** Global distribution of seagrass meadows (green) overlaid within six seagrass bioregions. Seagrass distribution data sourced from UNEP-WCMC & Short, (2018). Seagrass bioregions adapted from Short, Carruthers, Dennison, & Waycott (2007).


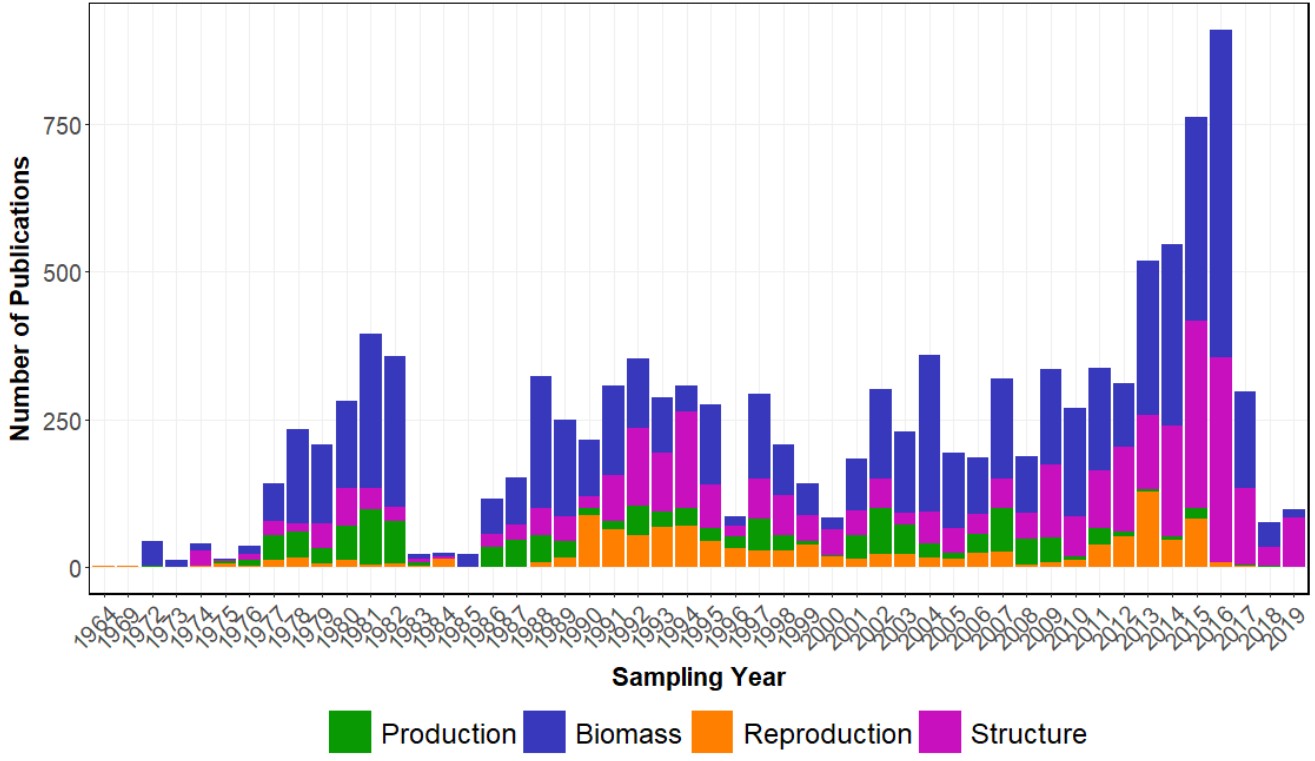

**Figure 2.** Number of publications that included seagrass data (coloured by type: biomass, structure, production and reproduction) based on the year of data collection. Data from peer-reviewed publications that did not report the year of sampling, were not included in this figure.

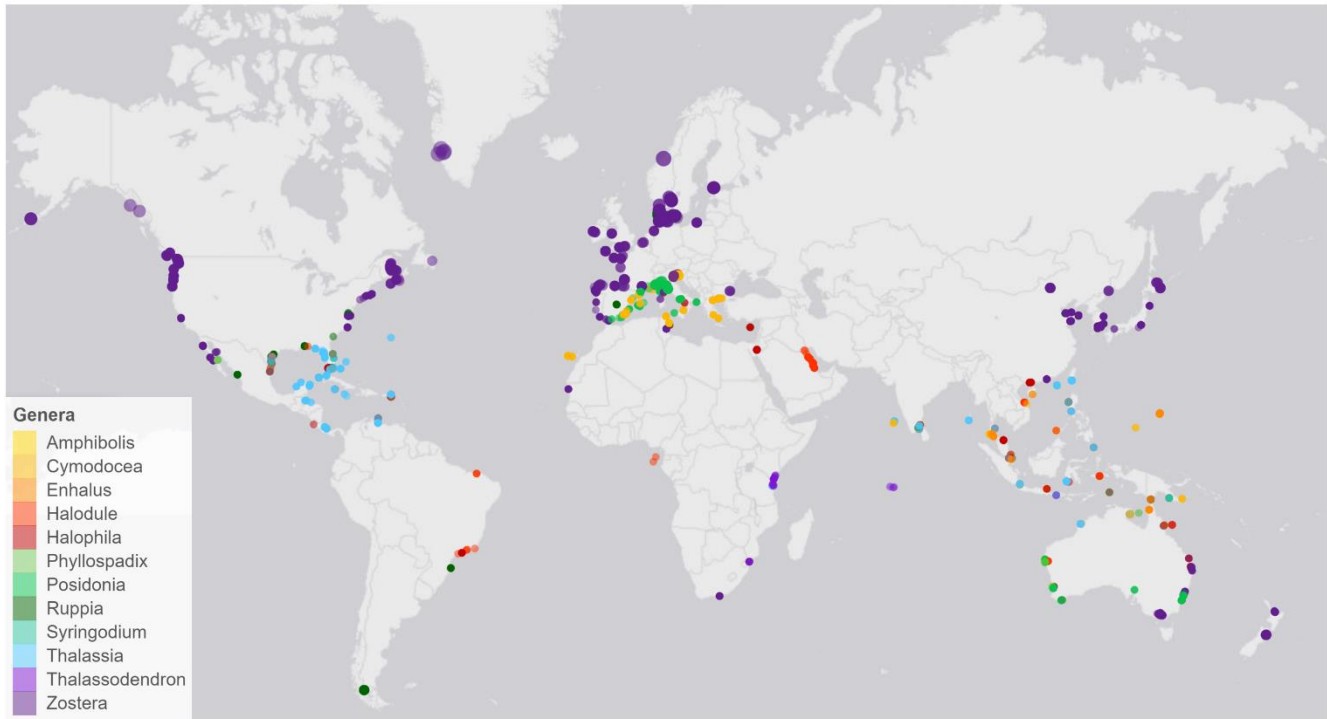

**Figure 3.** Global distribution map of data on seagrass structure, biomass, production and reproduction coloured by genera. The coloured points indicate the genera of seagrass studied and where many studies overlap, the colour appears darker than key.



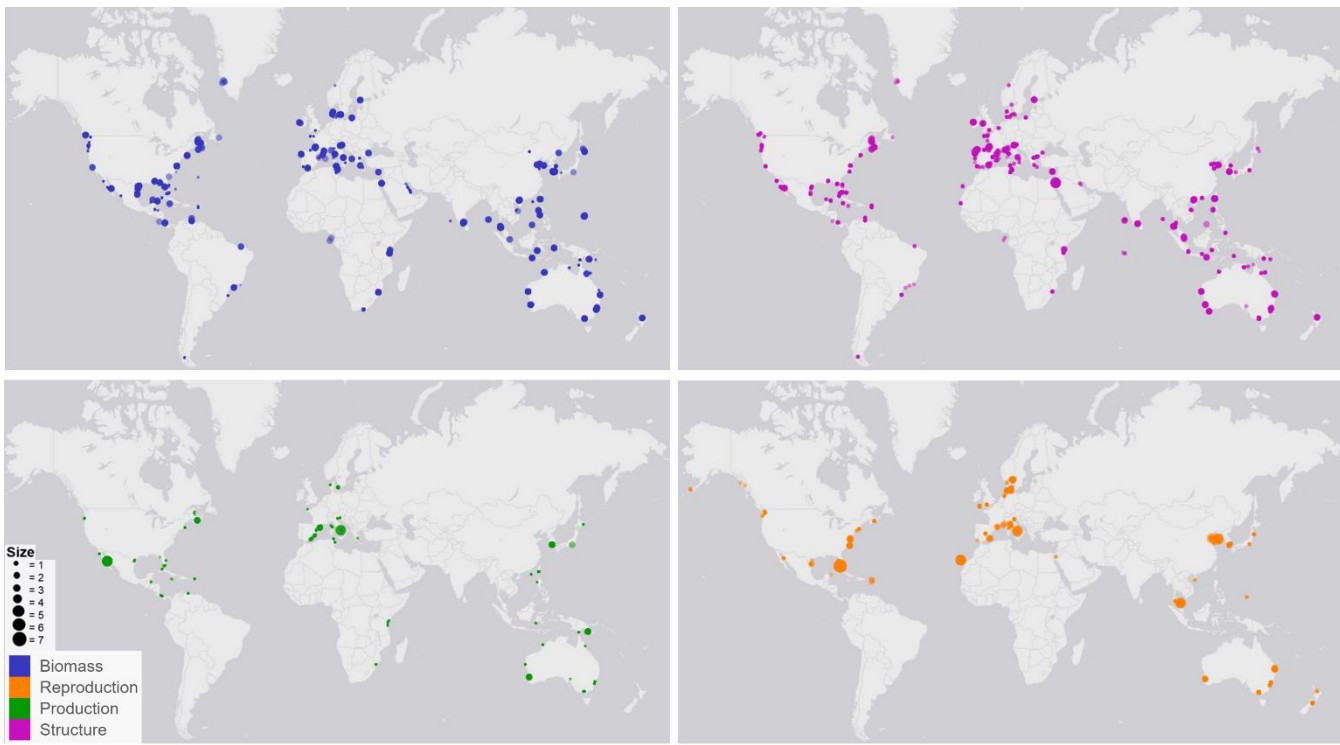


**Figure 4.** Global distribution map of seagrass study sites labelled as dots. The colours indicate the data type (biomass, reproduction, production and structure), while the size of each dot illustrates the number of data points for each site.



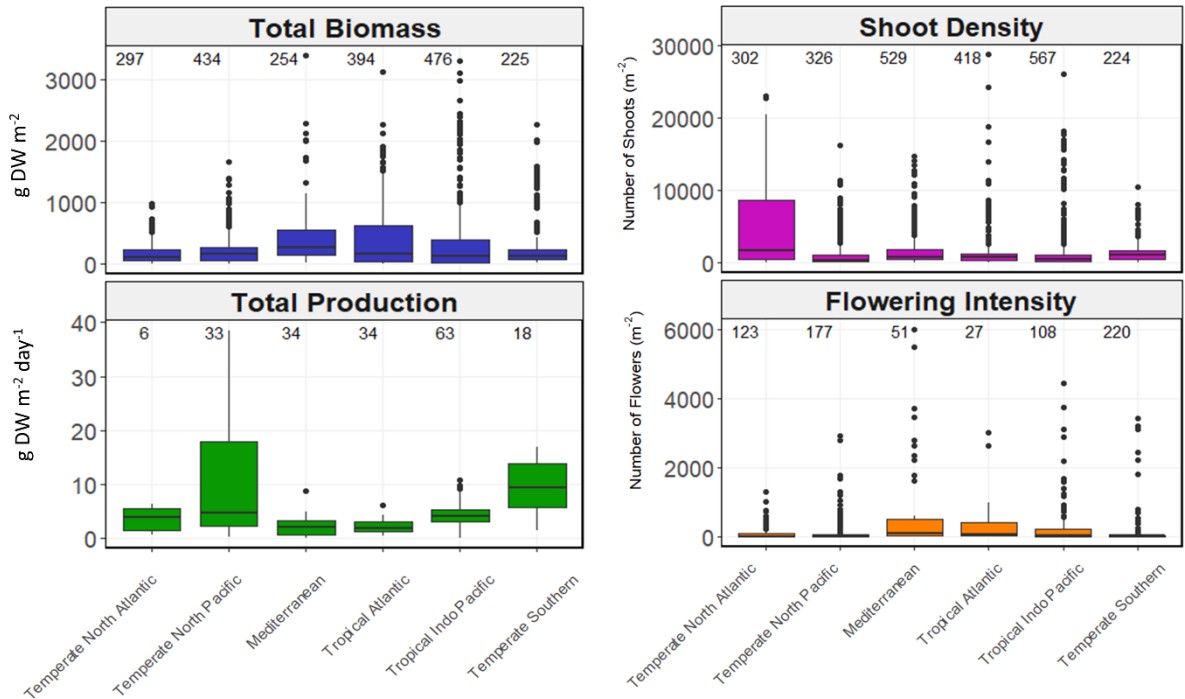

**Figure 5.** Box whisker plots depicting seagrass total biomass (including above-ground and below-ground biomass; g DW m$^{-2}$), shoot density (number of shoots m$^{-2}$), total production (g DW m$^{-2}$ day$^{-1}$) and flowering intensity (number of flowers or inflorescence shoots m$^{-2}$) values within each bioregion. The boxplots show the median value (black line within box), 75% and 25% percentiles create the top and bottom of the box and the tails are the maximum and minimum contributions within 1.5 interquartile range. Count of data (N) per bioregion is shown at the top of each whisker.



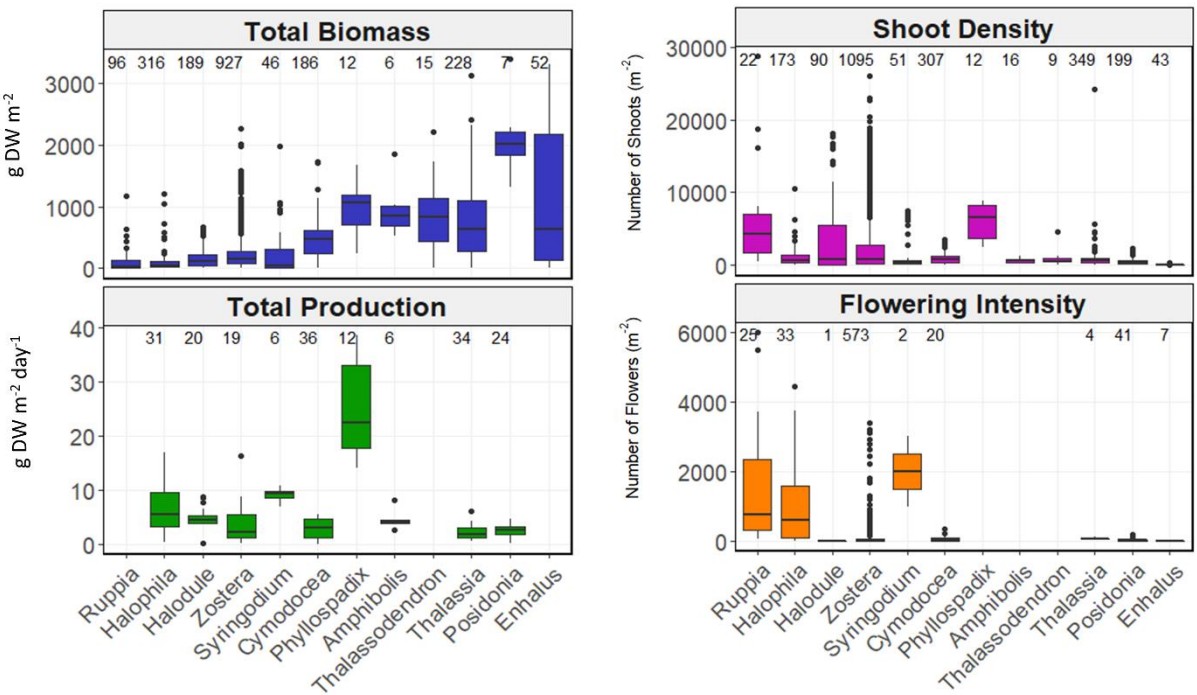

**Figure 6.** Box whisker plots depicting total biomass (g DW m$^{-2}$), shoot density (number of shoots m$^{-2}$), total net primary production (g DW m$^{-2}$ day$^{-1}$) and flowering intensity (number of flowers or inflorescence shoots m$^{-2}$) values per genera. The boxplots show the median value (black line within box), 75% and 25% percentiles create the top and bottom of the box and the tails are the maximum and minimum contributions within 1.5 interquartile range. Count of data (N) per bioregion is shown at the top of each whisker.






**Table 1.** Summary table outlining the count of data for each of the 15 seagrass variables based on bioregion, geotype and genera categorical variables. Above-ground biomass (AG), below-ground biomass (BG), total biomass (TB), shoot density (ShD), leaf density (LD), percent cover (Cov), above-ground production (AGP), below-ground production (BGP), total production (TP), shoot production (SP), leaf production (LP), flowering intensity (FI), fruit density (FD), sediment seed bank density (SB), seedling density (SD).

| Bioregion | Geotype | Genera | AG | BG | TB | ShD | LD | Cov | AGP | BGP | TP | SP | LP | FI | FD | SB | SD |
|---|---|---|---|---|---|---|---|---|---|---|---|---|---|---|---|---|---|
| Mediterranean | | *Cymodocea* | 12 | | 11 | 16 | | | | | | | | | | | |
| | Coastal | | 6 | 86 | 2 | 3 | 12 | 4 | 12 | 4 | 13 | | 82 | 20 | 23 | 17 | 21 |
| | | *Halophila* | 16 | 16 | 16 | 24 | 12 | 16 | | | | | | 2 | | | |
| | | *Posidonia* | 57 | 7 | 7 | 97 | 11 | 31 | 8 | 7 | 18 | 15 | 43 | 3 | 2 | | 22 |
| | | | | | | 11 | | | | | | | | | | | |
| | | *Zostera* | 79 | 20 | 33 | 3 | 24 | 9 | | | 3 | | 27 | 10 | | | 3 |
| | Estuarine | *Cymodocea* | 24 | 24 | 24 | 24 | | | 12 | | | | 12 | | | | |
| | | *Ruppia* | | | 2 | | | | | | | | | 16 | 2 | | |
| | | | | | | 10 | | | | | | | | | | | |
| | | *Zostera* | 66 | 49 | 60 | 8 | | 35 | | | | | | | | 1 | |
| Temperate North Atlantic | Coastal | *Halodule* | | | | | | 3 | | | | | | | | | |
| | | *Zostera* | 71 | 51 | 56 | 61 | 6 | 39 | 24 | 23 | | | 6 | 33 | | 25 | 8 |
| | Estuarine | *Ruppia* | 26 | 26 | 26 | | | 11 | | | | | | 8 | | 8 | |
| | | | 25 | 21 | 21 | 24 | | 12 | | | | | | | | | |
| | | *Zostera* | 2 | 5 | 5 | 1 | 12 | 9 | 14 | 6 | 6 | | 33 | 82 | | 82 | 11 |
| Temperate North Pacific | Coastal | *Halophila* | 23 | 23 | 42 | 19 | | | 13 | 13 | 13 | | | | | | |
| | | *Phyllospadix* | 12 | 12 | 12 | 12 | | 18 | 12 | 12 | 12 | 12 | 12 | | | | |
| | | | 12 | 13 | 20 | 18 | | | | | | | 15 | | | | |
| | | *Zostera* | 3 | 7 | 7 | 8 | 27 | 17 | | 1 | 1 | | 64 | 3 | | 11 | 11 |
| | Estuarine | *Ruppia* | 6 | | | 5 | | | | | | | | | | | |
| | | | 15 | 11 | 17 | 10 | | | | | | | | | | | |
| | | *Zostera* | 3 | 4 | 3 | 2 | 15 | | | | 7 | | 35 | 24 | | 53 | 12 |
| Temperate Southern | Coastal | *Amphibolis* | 2 | | 1 | 6 | 1 | | 1 | | | | | | | | |
| | | *Halophila* | | | | 3 | | | | | | | | | | | |
| | | | 12 | | | | | | | | | | | | | | |
| | | *Posidonia* | 4 | | | 49 | 8 | 24 | 9 | | | 13 | 11 | 3 | 2 | 3 | |
| | | *Zostera* | | | | | | | | | | | | 80 | | 1 | |



| Region | Habitat | Genus | 1 | 2 | 3 | 4 | 5 | 6 | 7 | 8 | 9 | 10 | 11 | 12 | 13 | 14 |
|---|---|---|---|---|---|---|---|---|---|---|---|---|---|---|---|---|
|  | Estuarine |  | 12 | 12 | 12 |  |  |  |  |  |  |  |  |  |  |  |
|  |  | *Halophila* | 5 | 5 | 5 |  |  |  | 6 | 6 | 18 |  |  |  |  |  |
|  |  | *Posidonia* |  |  |  | 40 |  |  |  |  |  | 37 | 35 |  |  | 4 |
|  |  | *Ruppia* |  |  | 11 | 5 |  |  |  |  |  |  |  |  |  |  |
|  |  |  | 15 |  |  | 12 | 16 |  |  |  |  |  | 10 |  |  |  |
|  |  | *Zostera* | 9 | 60 | 88 | 1 | 7 | 55 |  |  |  | 2 | 2 | 2 |  |  |
| Tropical Atlantic | Coastal | *Halodule* | 18 | 8 | 11 | 27 |  |  |  |  |  |  | 1 | 1 | 11 |  |
|  |  | *Halophila* | 9 | 9 | 38 | 19 | 1 |  |  |  |  |  | 9 | 9 | 8 | 10 |
|  |  | *Syringodium* | 5 | 5 | 21 | 7 |  |  |  |  |  |  | 1 | 2 | 1 |  |
|  |  |  |  |  | 11 | 19 |  |  |  |  |  |  |  |  |  |  |
|  |  | *Thalassia* | 87 | 51 | 2 | 8 | 1 | 10 |  |  | 33 | 10 | 13 | 4 | 3 | 2 |
|  |  | *Zostera* | 3 | 3 | 3 | 9 | 6 |  |  |  |  | 24 | 24 | 4 |  |  |
|  | Estuarine | *Halodule* | 30 | 30 | 35 | 6 | 7 |  |  |  |  |  |  |  |  |  |
|  |  | *Halophila* |  |  | 3 |  | 2 |  |  |  |  |  |  |  |  |  |
|  |  | *Ruppia* | 48 | 52 | 57 | 12 | 14 |  |  |  |  |  |  | 1 |  |  |
|  |  | *Syringodium* | 1 | 2 | 7 | 5 | 2 |  |  |  |  |  | 5 |  |  |  |
|  |  | *Thalassia* | 26 | 21 | 46 | 33 | 2 | 2 |  |  | 1 |  | 32 |  |  |  |
|  |  |  |  |  |  | 10 |  |  |  |  |  |  |  |  |  |  |
|  |  | *Zostera* | 44 | 13 | 61 | 2 |  |  |  |  | 25 |  | 6 |  |  |  |
| Tropical Indo Pacific | Coastal | *Amphibolis* | 16 |  | 5 | 10 | 5 |  | 5 |  | 6 |  |  |  |  |  |
|  |  | *Cymodocea* | 12 |  |  | 10 |  |  |  |  |  |  |  |  |  |  |
|  |  |  | 0 | 34 | 48 | 7 |  | 22 | 2 | 10 | 21 | 2 | 34 |  | 2 |  |
|  |  | *Enhalus* | 65 | 39 | 41 | 40 |  | 17 | 1 |  |  |  | 11 |  |  |  |
|  |  |  |  |  | 11 |  |  |  |  |  |  |  |  |  |  |  |
|  |  | *Halodule* | 84 | 60 | 6 | 51 |  | 12 | 1 |  | 18 | 1 | 22 |  |  |  |
|  |  |  | 11 |  |  |  |  |  |  |  |  |  |  |  |  |  |
|  |  | *Halophila* | 0 | 68 | 73 | 90 | 1 | 46 | 3 | 1 |  |  |  |  |  |  |
|  |  | *Posidonia* | 6 |  |  | 13 |  |  |  |  | 6 |  |  |  |  |  |
|  |  | *Syringodium* | 48 | 17 | 18 | 39 |  | 5 |  | 6 | 6 |  | 6 |  |  |  |
|  |  |  | 22 |  |  | 10 |  |  |  |  |  |  | 14 |  |  |  |
|  |  | *Thalassia* | 6 | 57 | 68 | 9 |  | 43 | 1 |  |  | 2 | 4 |  |  |  |
|  |  | *Thalassodendron* | 24 | 12 | 13 | 7 |  | 5 | 1 |  |  |  | 11 |  |  |  |



| | | | | | | | | | | | | | | | | |
|---|---|---|---|---|---|---|---|---|---|---|---|---|---|---|---|---|
| Estuarine | Zostera | 66 | 9 | 10 | 6 | 4 | 20 | | | | 1 | | 60 | | 80 | |
| | Cymodocea | 5 | 2 | 2 | 13 | | | | | 2 | | | | | | |
| | Enhalus | 3 | | 11 | 3 | | 2 | | | | | | 7 | | | |
| | Halodule | 17 | 17 | 27 | 6 | | 11 | | | 2 | | | | | | |
| | Halophila | 4 | 4 | 19 | 18 | | 19 | | | | | | 22 | 11 | | 4 |
| | Thalassia | 1 | | 2 | 9 | | 2 | | | | | | | | | |
| | Thalassodendron | 2 | 3 | 2 | 2 | | | | | | 3 | 2 | | | | |
| | Zostera | 7 | 7 | 21 | 44 | | 10 | | | 2 | | | 19 | | 12 | |
| **Total # data** | | **2519** | **1488** | **2080** | **2366** | **159** | **731** | **192** | **89** | **188** | **110** | **670** | **706** | **55** | **312** | **108** |