# Peer review of "Global dataset on seagrass meadow structure, biomass and production"

_Earth System Science Data, 2022_

## Author Response (AR1)

Dear Editor,

We are pleased to resubmit our manuscript *'Global dataset on seagrass meadow structure, biomass and production'* to the Earth System Science Data journal. We thank the reviewers for their comments, which have helped us to strengthen our article. In addition to the comments received by the reviewers, we have omitted the reproduction data from the manuscript given the lack of time to review this part of the dataset. The removal of the reproduction data does not significantly impact the usefulness of our review to advance science, owing to the large number of variables extracted in regards to seagrass structure, biomass and production. After the data curation, the total number of data increased from 11,773 to 14,271.

We have addressed all the suggestions made by the reviewers throughout the manuscript. We hope these revisions are to your satisfaction and thank you for considering our article for publication in the Earth System Science Data journal.

Sincerely,

Dr Simone Strydom
(On behalf of all co-authors)

Dr Oscar Serrano

**Response to Reviewers essd-2022-160**

Title: Global dataset on seagrass meadow structure, biomass and production
Author(s): Simone Strydom et al.
MS No.: essd-2022-160
MS type: Data description paper

| Reviewer (date comment published) | Comment | Response (date posted) |
|---|---|---|
| **CC** (community comment) Albert Pessarrodon a Silvestre (08/08/2022) | Wonderful resource. Would be great to include some sort of metadata on when the measurements were taken given the well-established relationship between season and some of the variables (e.g., productivity). A similar output has just been published for macroalgae, which may bring us closer to examining the production of submerged vegetated habitats! | Thank you, Albert, for your positive feedback. Data on seasonality was initially sought during the review process, however it become apparent that this was not going to be possible because many articles did not report the date of data collection, therefore we could not be sure of season. Instead, we focused on categorical variables that could be included for the majority of numerical variables compiled. For example, the dataset reports the year of data collection which could be useful for deciphering long-term temporal trends in seagrass structure, biomass, net primary production and reproduction variables, even if an effect of season cannot be delineated. (10/08/2022) |
| **RC1** (01/09/2022) | This paper presents an interesting method to collate seagrass records from around the globe. However, the search terms are a little too restrictive and there are useful papers that don't seem to have been included. For examples a quick search using different search engine rather than just WOS finds other papers with suitable records: e.g. https://www.sciencedirect.com/science/article/abs/pii/S1470160X17303783 & https://www.mdpi.com/2077-1312/8/9/717 Also some grey-literature records could provide useful information, e.g.: https://publications.jrc.ec.europa.eu/repository/handle/JRC115082 The fact that a very quick internet search can find missing useful information suggest that a more comprehensive data search would greatly improve the usefulness of this work. The links to the database did not work in the preprint supplied so this has not been reviewed. | We concur that the search terms used were specific but were thoroughly chosen by a group of seagrass experts who concluded that there are the most widely used terms (e.g., keywords) by seagrass researchers across the literature. The search terms used returned ~7,000 publications in WOS that were reviewed carefully to decipher which ones contained useful data for our review (18% of the total number of manuscripts depicted by WOS). Prior to deciding which terms to use, we run several searches with multiple terms to aid the decision of which terms to use. Adding new terms to the search overall returned hundreds of additional publications with a very low percentage of suitable manuscripts for our review (<1%) and thereby, adding unnecessary complexity to our review. Similarly, we have chosen to use WOS instead of other search engines based on previous studies showing the suitability of this engine for conducting systematic reviews and meta-analyses (e.g., Gusenbauer and Haddaway, 2019). We included a sentence in the manuscript acknowledging that our search has likely missed a small portion of the peer-reviewed data published, owing to the use of different terms across research fields, and the use of a single search engine WOS to conduct the review.

We thank the reviewer for the links and suggestions. However, Wilkes et al 2017 did not appear to contain data that fit our criteria for extraction because it only reports data on change in seagrass spatial extent which was not included in our data review (i.e., we only included the following data on meadow structure: % cover, shoot density and leaf density). The second link (Kletou et al. 2020) was published in September 2020 that was beyond the search time point |

| | | | for this work (listed in the Methods section as June 2020) and therefore it did not appear in our search. We do not have the capacity to extend the search period at this stage. |
|---|---|---|---|
| | | | We focussed our review on peer-reviewed literature to ensure high quality of the data compiled. Although the grey literature may contain relevant data, it was not possible to assess their robustness and was not considered. |
| | | | In this re-submission, we have provided the excel file with the whole dataset to allow a thorough review. Note that the dataset in Pangaea is under embargo until the publication of this manuscript. |
| **RC2** (01/09/2022) | This data set and its description is useful, and I can foresee its use in many different ways. My specific comments are as follows:

1. In the introductory paragraph about seagrass ecosystem functions and services, consider adding water purification/filtration to the list. Suggested references: Lamb et al 2017 (Science), Ascioti et al 2022 (Ecosystem Services).
2. In section 2.1 (literature search), the stated search terms did not include *Syringodium*, but this species was in the results. The term 'exten' in the search - should it be 'exten' or 'extent'?
3. In Section 2.2, the last paragraph (line 120) seemed out of place because it described the way natural history reporting has evolved, not a method. You may want to consider moving this to the Results and Discussion section.
4. I found it difficult to differentiate between species because of the colour gradient in Figure 3 - the yellows/oranges in particular (Amphibolis, Cymodocea, Enhalus), were harder to make out than the rest. On this note, I'd suggest checking for the use of colorblind safe gradients in ColorBrewer (https://colorbrewer2.org/#type=sequential&scheme=BuGn&n=3).
I think this map is useful for summarizing research hotspots and gaps at a glance, and it would be a shame if the reader did not get the full experience of it.
5. Section 4 (Line 196): there is a mismatch between the text and abstract. The text says, "...the least number of data was related to seagrass reproduction (9% of data)" but the abstract says it's production that has the least data points, at 10%.
6. Nice work in building this data set - this was a tremendous effort. I did notice some missing papers. In many of the papers with such seagrass data, the titles and keywords often don't use the search terms you've selected. We often use | | We would like to thank the reviewer for the helpful suggestions and for acknowledging the tremendous effort this dataset took to produce.
1. The suggested references have been added to the introductory paragraph (line 48).
2. Unfortunately, the omission of *Syringodium* was an error so we re-run the search to fill any publications that we may have missed and added them. Note that there were already 213 entries for *Syringodium* from other papers that were picked up from other search terms. In Section 2.1, 'exten' was a deliberate choice to include 'extension' and 'extent' without doubling up.
3. The last paragraph about natural history reporting has been removed from Section 2.2, shortened and the key point included in the Conclusions section (line 213).
4. Thank you for the suggestions for colour palettes. We struggled with this as having 12 colours easily distinguishable and colourblind friendly was complicated. We tried the Set 3 pastel gradient in R's ColorBrewer (as suggested by the Reviewer) but it is still hard to tell some of the genera apart. We have now updated Figure 3 to manually alter colours to a more suitable palette.
5. In Section 4 the values mismatch between Production vs Reproduction (9% vs 10%) have been corrected. However, note that the values changed in the new version submitted due to the new data included.
6. We appreciate the papers suggested, they have been examined and the appropriate data added to the database. Except for McKenzie et al 2016 which had data collated for multiple species into colonising/opportunistic/ persistent groups that could not be merged into our dataset because it reports specific data for genera rather than for life-trait groups. |

terms such as 'condition' or 'status', so this is possibly why some papers were not picked up in your search. Here are some additional papers that have the data you're after but are not in your list:

- Marba, N., Duarte, C. M., Terrados, J., Halun, Z., Gacia, E., & Fortes, M. D. (2010). Effects of seagrass rhizospheres on sediment redox conditions in SE Asian coastal ecosystems. Estuaries and Coasts, 33(1), 107-117. doi:10.1007/s12237-009-9250-0
- McKenzie, L. J., Yaakub, S. M., Tan, R., Seymour, J., & Yoshida, R. L. (2016). Seagrass habitats of Singapore: Environmental drivers and key processes. [ENV REQ]. Raffles Bulletin of Zoology(34), 60-77.
- Muta Harah, Z., Japar Sidik, B., & Hishamuddin, O. (1999). Flowering, fruiting and seedling of Halophila beccarii Aschers. (Hydrocharitaceae) from Malaysia. Aquatic Botany, 65(1-4), 199-207.
- Novak, A. B., Hines, E., Kwan, D., Parr, L., Tun, M. T., Win, H., & Short, F. T. (2009). Revised ranges of seagrass species in the Myeik Archipelago, Myanmar. Aquatic Botany, 91(3), 250-252. doi:10.1016/j.aquabot.2009.07.002
- Ooi, J. L. S., Kendrick, G. A., Van Niel, K. P., & Affendi, Y. A. (2011). Knowledge gaps in tropical Southeast Asian seagrass systems. Estuarine, Coastal and Shelf Science 92(1), 118-131. doi:doi:10.1016/j.ecss.2010.12.021
- Terrados, J., Duarte, C. M., Fortes, M. D., Borum, J., Agawin, N. S. R., Bach, S., . . . Vermaat, J. (1998). Changes in community structure and biomass of seagrass communities along gradients of siltation in SE Asia. Estuarine Coastal and Shelf Science, 46(5), 757-768.